



# Turbulence Detection in the Atmospheric Boundary Layer using Coherent Doppler Wind Lidar and Microwave Radiometer

Pu Jiang[1], Jinlong Yuan[1], Kenan Wu[1], Lu Wang[1], and Haiyun Xia[1,2,3]

[1]School of Earth and Space Science, University of Science and Technology of China, Hefei 230026, China
[2]Hefei National Laboratory for Physical Sciences at the Microscale, University of Science and Technology of China, Hefei 230026, China
[3]CAS Center for Excellence in Comparative Planetology, Hefei 230026, China

*Correspondence to*: Haiyun Xia (hsia@ustc.edu.cn)

**Abstract.** The refractive index structure constant ($C_n^2$) is a key parameter in describing the influence of turbulence on laser
transmission in the atmosphere. A new method for continuous $C_n^2$ profiling with both high temporal and spatial resolution is proposed and demonstrated. Under the assumption of the Kolmogorov "2/3 law", the $C_n^2$ profile can be calculated by using the wind field and turbulent kinetic energy dissipation rate (TKEDR) measured by coherent Doppler wind lidar (CDWL) and other meteorological parameters derived from microwave radiometer (MWR). In the horizontal experiment, a comparison between the results from our new method and measurements made by a large aperture scintillometer (LAS) is conducted. Except for
the period of stratification stabilizing, the correlation coefficient between them in the six-day observation is 0.8389, the mean error and standard deviation is $1.09 \times 10^{-15}$ m$^{-2/3}$ and $2.14 \times 10^{-15}$ m$^{-2/3}$, respectively. In the vertical direction, the continuous observation results of $C_n^2$ and other turbulence parameter profiles in the atmospheric boundary layer (ABL) are retrieved. More details of the atmospheric turbulence can be found in the ABL owe to the high temporal and spatial resolution of MWR and CDWL (spatial resolution of 26 m, temporal resolution of 147 s).

## 1 Introduction

Turbulence analysis is meaningful in many fields, such as astronomy (Ma et al., 2020), aviation safety (Storer et al., 2019), optical communication technology (Ren et al., 2013), laser weapons (Extance, 2015), wind field retrieval (Kumer et al., 2016), air pollution (Wei et al., 2020b), oceanography (Nootz et al., 2016), etc. In astronomy, due to the existence of atmospheric turbulence, the star images seen in astronomical observations often jitter and flicker, so that increasing the aperture of the
telescope in ground-based observations cannot achieve the expected results. To reduce the effects, researchers need to seek for sites with high altitudes or places like Dome A in Antarctica (Ma et al., 2020), where the atmospheric boundary layer (ABL) is thin. In addition, in areas of Free-Space Optical (FSO) communications and laser ranging, the fluctuation of the refractive index caused by atmospheric turbulence will affect the coherence of the laser beam through the optical angle-of-arrival fluctuation, laser beam wander and scintillation (Libich et al., 2017), etc. These phenomena will bring uncertain variance and
reduce the detection efficiency of these systems.



There are several parameters normally used to describe turbulence, for example, the Reynolds number, turbulence kinetic energy, Richardson number, refractive index structure constant, etc. Among all these parameters, the refractive index structure constant $C_n^2$ is the one that most directly represents the fluctuation intensity of the spatial refractive index, which makes it vital especially in optical turbulence. At present, there are some common methods for $C_n^2$ detection. For instance, there are methods derived from astronomy and satellite-ground laser communication areas, such as the Slope Detection and Ranging (SLODAR) (Butterley et al., 2006), Coupled Slope and Scintillation Detection and Ranging (CO-SLIDAR) (Voyez et al., 2014), etc. (Fusco and Costille, 2010; Otoniel Canuet, 2015; Voyez et al., 2012). The empirical models represent the average characteristics of different regions, such as the Hufnagel-Valley (HV) profile model (Andrews et al., 2009; Beland, 1993; Chen et al., 2016; Wu et al., 2021). Through the Radiosonde and sounding balloon, turbulence profile can be acquired with a high spatial resolution (Barletti et al., 1974; He et al., 2020; Ko et al., 2019; Martini et al., 2017; Zhang et al., 2019). There are some methods normally applied to the near-surface, which use the relationship between $C_n^2$ and temperature fluctuation (Odintsov et al., 2019; van Iersel et al., 2019; Vyhnalek, 2017). Large aperture scintillometer (LAS) can measure the path-averaged $C_n^2$ within a certain distance with high time resolution, which is a common instrument using the intensity scintillation principle (Andrews et al., 2012; Han et al., 2018; Ting-i et al., 1978). Through adjusting the focal length and remaining at each height to reduce the variance of the distribution of lidar profiles, imaging methods that use the differential image motion monitor (DIMM) are also quite mature way to detect the turbulence intensity (Aristidi et al., 2019; Belen'kii et al., 2001; Brown et al., 2013; Chabe et al., 2020; Cheng et al., 2017; Gimmestad et al., 2012; Jing et al., 2013). Atmospheric backscattering amplification method by measuring the intensity of laser echo signal amplification effect (Banakh and Razenkov, 2016a, b; Razenkov, 2018), etc. However, there are some limitations of these methods in effectively detecting the rapidly changing atmospheric environment. For example, it normally takes a long time using balloon methods (Barletti et al., 1974), the saturation of optical scintillation in long distance, it is difficult for imaging methods to improve spatial resolution (Gimmestad et al., 2012), the response to temperature fluctuation is not sensitive enough using the Raman lidar (Whiteman et al., 2011). Consequently, it is necessary to seek for a method that can detect the turbulence profiles with high temporal and spatial resolution at the same time.

Coherent Doppler wind lidar (CDWL) obtains radial wind speed information by retrieving the frequency shift of the received signal after passing through a distance of atmosphere (Shangguan et al., 2017; Wang et al., 2019b). Its reliable performance has been verified in various areas, such as: turbulence parameters (Banakh et al., 2017; Banakh et al., 2020a, 2021a; Banakh et al., 2021b; Smalikho and Banakh, 2020), aircraft wake vortices (Smalikho et al., 2020), boundary layer height (Wang et al., 2019a; Wang et al., 2021; Yang et al., 2020), gravity waves (Cao et al., 2016; Jia et al., 2019), low-level jets (LLJs) (Banakh and Smalikho, 2018; Banakh et al., 2020a; Tuononen et al., 2017), simultaneous wind and rainfall detection (Wei et al., 2019), identifying different atmospheric environments (Yuan et al., 2020) and others (Wang et al., 2017; Wei et al., 2020a). Therefore, using CDWL is a promising method to achieve accurate and high-resolution turbulence detection.

This paper begins with the principle to estimate the $C_n^2$ in Section 2. In Section 3, a horizontal comparison experiment with LAS is carried out, then the continuous observation results of $C_n^2$ profiles are retrieved vertically. Later, several key parameters are analyzed to study the turbulence characteristics. After that, five typical sets of 12-min continuous turbulent profiles





represent different periods of daytime and nighttime are analyzed and compared with the turbulence model. Finally, conclusions are given in Section 4.

## 2 Principle

According to Tartarski's theory (Tatarskii, 1961), by combining the outer scale of turbulence $L_0$ or the wind field data with meteorological parameters, $C_n^2$ can be calculated by equation (1) (Phillips, 2005):

$$C_n^2 = a^2 L_0^{4/3} M^2 = a^2 \times \left[ \frac{K^2}{\left(\frac{\partial U}{\partial z}\right)^2} \right]^{1/3} \times M^2 , \qquad (1)$$

where $a^2$ is a constant, usually takes the experimental empirical value of 2.8 (E. M. Dewan, 1993), and $M$ is the vertical gradient of generalized potential refractive index. $K$ is the turbulence dissipation coefficient, $\partial U / \partial z$ is the wind shear of the horizontal wind vector in the vertical direction. In the former method, the calculation of $L_0$ is quite difficult, usually it is estimated by different outer scale models (Coulman et al., 1988; Han et al., 2020), such as the C-V model (Coulman et al.,

1988), Dewan model (E. M. Dewan, 1993), etc. For example, the Dewan model has the expression:

$$L_0^{4/3}(z) = \begin{cases} 0.1^{4/3} \times 10^{1.64+42 \times S}, troposphere \\ 0.1^{4/3} \times 10^{0.506+50 \times S}, stratosphere \end{cases}, \qquad (2)$$

where $z$ is the height, and $S$ is the intensity of the wind shear.

Most of these models are related to height, wind shear intensity and temperature gradient. Empirical models are normally obtained from long-term statistical averaged data with an in-situ sounding balloon, which is hard to reflect the local

atmospheric features in different areas. In addition, wind shear and temperature gradient cannot be measured directly, they are closely related to the scale chosen for calculation. It would be more robust when choosing a larger scale, but might lose the detailed messages, and would be more sensitive on the contrary. In order to reduce the error, it is necessary to avoid using these parameters to calculate as much as possible.

In the "inertial sub-region" (between inner scale $l_0$ and outer scale $L_0$), where the refractive index structure function obeys the

Kolmogorov "2/3 law" (Kolmogorov, 1962), the turbulent energy propagates from the outer scale to the inner scale without dissipation. There is a relationship between $K$ and the turbulent kinetic energy dissipation rate (TKEDR) $\varepsilon$:

$$\varepsilon = K \left(\frac{\partial U}{\partial z}\right)^2 , \qquad (3)$$

substituting Eq. (3) into Eq. (1) yields:

$$C_n^2 = a^2 \times \frac{\varepsilon^{2/3}}{\left(\frac{\partial U}{\partial z}\right)^2} \times M^2 , \qquad (4)$$



TKEDR can be calculated by matching the measured radial velocity and azimuth structure function with the theoretical structure function (Banakh and Smalikho, 2018). In addition, $\partial U / \partial z$ can be calculated by the following equation:

$$\left(\frac{\partial U}{\partial z}\right)^2 = \left(\frac{\partial u}{\partial z}\right)^2 + \left(\frac{\partial v}{\partial z}\right)^2 , \tag{5}$$

where $u$ and $v$ are the zonal wind and meridional wind, respectively. And $M$ has the expression of (E. M. Dewan, 1993):

$$M = -79 \times 10^{-6} \frac{P}{T} \left(\frac{\partial \ln \theta}{\partial z}\right) , \tag{6}$$

where $\theta$ is the potential temperature:

$$\theta = T \left(\frac{1000}{P}\right)^{0.286} , \tag{7}$$

where $T$ represents temperature (K). $P$ means pressure (hPa). In order to simplify the calculation and analyse the effects of temperature and pressure gradients, merging the Eq. (6) and Eq. (7) yields:

$$M = -79 \times 10^{-6} \times \frac{P}{T} \times \left(\frac{1}{T}\frac{dT}{dz} - \frac{0.286}{P}\frac{dP}{dz}\right) . \tag{8}$$

From Eq. (4) to Eq. (8), one can see that, with the TKEDR and wind profiles from CDWL and temperature and pressure profiles from microwave radiometer, the refractive index structure constant $C_n^2$ can be estimated.

## 3 Experiments

### 3.1 Instruments

The CDWL applied in this paper (Site A of Fig. 1) uses an eye-safe 1.55 μm wavelength in the transmitting system, the single
pulse energy of the laser is 300 μJ, and the repetition frequency is 10 kHz. In the experiment, the CDWL adopts the velocity azimuth display (VAD) scanning mode, the elevation angle is fixed at 60 degrees. The step length of the scanning azimuth angle is 5 degrees, and the scanning period is 147 s, which is the same as the temporal resolution of the retrieved wind field. In the vertical direction of 0-2.17 km, 2.17-4.76 km, and 4.76-11.26 km, the range resolution is 26 m, 52 m, and 130 m, respectively. Besides, the lidar adopts an all-fiber structure and temperature control system to ensure stability. Specific
parameters of the CDWL are listed in Table 1.

The ground-based MWR used in the experiment has a time resolution of 2 min. In the vertical direction of 0-0.5 km, 0.5-2 km, and 2-10 km, the distance resolution is: 50 m, 100 m, and 250 m, respectively. Ground surface measurements contain temperature, humidity, pressure, cloud base height, and vertical observation results can provide profiles of temperature, water vapor density, relative humidity, and liquid water content within 0-10 km. There are more specific introductions and
115 applications about MWR in (Pan et al., 2020).

Under the assumption of horizontally homogeneous, LAS (Kipp&Zonen LAS MKII, Fig. 1.) can retrieve the path-averaged $C_n^2$ by detecting the light intensity fluctuations of signals in the receiving end that passing through a distance of atmosphere.



LAS is chosen for the verification experiment due to its mature theory and reliable results. It can provide the $C_n^2$ information at the temporal resolution of 1 s. In this experiment, 10 min time interval is adopted to reduce the data fluctuation.

**3.2 Verification experiment**

A horizontal direction verification experiment is carried out on the University of Science and Technology of China (USTC) campus in Hefei, Anhui Province (31°50′10″ N, 117°16′10″ E). As shown in Fig. 1, the CDWL and the DAVIS weather station are placed at site A. The receiving and transmitting ends of LAS are located at the height of 55 m at site A and site B respectively. Its detection path (in the direction of the red arrow) is a north-south direction to avoid the influence of sunlight.

The two sites are 900 meters apart, which is within the best detection range of LAS. The wind tower is placed at site C to record the continuous data of temperature and for the calculation of temperature gradient.

As shown in Fig. 2, six days of continuous observations are carried out from 00:00 on September 26 to 24:00 on October 01, 2020, local time. Fig. 2(a)-(e) represent the wind field and TKEDR results obtained from the CDWL, Fig. 2(f) is the temperature data recorded at the height of 2m, 8m, and 18m of the wind tower and the temperature gradient result calculated

by linear fitting. Since the height of the wind tower is about 18 meters and near the ground, which represents the changes in a local area of the surface, so the calculated temperature gradient can vary widely every day. Therefore, to reduce the error, in the process of calculating $C_n^2$ in the horizontal experiment, the temperature gradient is taken as the empirical value in the troposphere: -0.0065 K/m, the pressure gradient is: -0.10 hPa/m. Then the results of $C_n^2$ measured from LAS and retrieved from CDWL combined with meteorological data are the red line and black dot-dash line in Figure 2(g), respectively.

It can be seen from Fig. 2(g) that the result retrieved by CDWL is generally higher than LAS during the period around 16:00-20:00, the $C_n^2$ measured by LAS usually declines rapidly to a minimum at sunset, while it decreases slowly until 20:00 when using CDWL.

Combining with the daily trend of the temperature gradient in Fig. 2(f), it can be found that it has a strong negative correlation with the $C_n^2$ result detected by LAS and the temperature gradient. Obviously, there is a strong temperature inversion in the

140 surface layer from afternoon to night every day. On the one hand, when the temperature gradient changes from negative to positive around 16:00 in the afternoon, the convection activities weaken instantly and the $C_n^2$ drops rapidly (Guo et al., 2020; Olofson et al., 2009). It can also be verified from the vertical wind speed in Fig. 2(c) that it becomes more stable and approaches zero when the temperature inversion develops.

On the other hand, with the temperature gradient growing, the stratified structure begins to form. It can be proved by the Brunt–

145 Väisälä frequency, which is a common way to estimate the stability of the atmosphere stratification (Balsley et al., 2018; Friedrich et al., 2012; Jia et al., 2019; Sorbjan and Balsley, 2008). The squares of the buoyancy frequency have the expression of:

$$N^2 = \frac{g}{\theta}\left(\frac{\partial \theta}{\partial z}\right),$$  (9)





where $g$ is the gravitational acceleration and $\theta$ is the potential temperature. When $N^2 > 0$, the stratification structure is stable,
and becomes unstable when $N^2 < 0$ due to the inversion of the density. The continuous-time profile of $N^2$ calculated from the
wind tower is shown in Fig. 2(g). It can be found that $N^2$ also turns from negative to positive after around 16:00 like the
temperature gradient, which means the atmosphere stratification becomes more stable and the laminar flow grows. As a result,
the anisotropic component of turbulence grows (Banakh and Smalikho, 2019).

While the atmospheric energy is still slowly dissipating in the horizontal direction, so the TKEDR near the surface is still quite
high as shown in Fig. 2(e), which makes the $C_n^2$ results retrieved from CDWL are higher. After the average temperature of the
atmosphere drops at night, the TKEDR also gradually decreases, then the $C_n^2$ results from the two instruments become
consistent again. However, different from the transition period in the afternoon, the stratification structure becomes unstable
and the turbulence intensity strengthens from the surface after the sunrise. Therefore, the $C_n^2$ calculated from these two methods
coincides with each other in the morning.

Besides, the LLJs can often be found at the night, especially on Sep. 26, 27, and 30. The occurrence of LLJ usually leads to an
increase in wind shear, which in turn breaks the stability of the atmosphere and generates turbulent flow. However, unlike the
daytime, the increase of wind shear did not cause a wide range of turbulence due to the suppressing effect of the Brunt–Väisälä
frequency. Instead, only small fluctuations occurred on the edge of the LLJ, which can be found in the slight variations of $C_n^2$
at night.

Except for the period of differences in the afternoon (about 16:00-20:00), the fluctuating trend and orders of magnitude of the
$C_n^2$ results in the rest of the time are quite consistent every day. The statistical analysis of the results of six-days continuous
observations from LAS and CDWL is shown in Fig. 3. The periods around 16:00-20:00 every day are plotted in colour circles
and all the rest are in black dots.

When using all data for analysis, the correlation coefficient and mean error between the two methods are 0.6723 and 1.34×10-
15 m-2/3, respectively. When using the black dots, the correlation coefficient, mean error, and standard deviation RMSE are
0.8389, 1.09×10-15 and 2.14×10-15 m-2/3, respectively. As a consequence, the method using CDWL is feasible except for the
transition period when the stratification becomes more stable.

### 3.3 Continuous observation of $C_n^2$ profile

Based on the results of the horizontal verification experiment, the results of the continuous field experiment conducted on the
Xilin Gol Prairie (43°54′ N, 115°58′ E) in Inner Mongolia in September 2019 are analyzed and discussed. Different from cities,
the topography of the grassland makes the atmosphere in the convective boundary layer (CBL) more meets the homogeneous
assumption, so the measured results are more representative for this area.

Figure 4(a)-(c), (e)-(g) show the wind field and TKEDR results retrieved from CDWL in two continuous days from 10:00 on
September 06 to 24:00 on September 07, 2019, local time. The quality of data is affected by carrier-to-noise (CNR), which is
the ratio of total signal power to the noise power, and it is the main parameter that determines the accuracy of wind field





retrieval. In this paper, the data with CNR above -35 dB are selected for retrieving radial wind speed (Wang et al., 2017). Due to the effect of turbulence, the boundary layer height is uplifted during the daytime, so the CNR can be detected higher accordingly. A sudden increase of CNR above 2.2 km is caused by the change in distance resolution. In addition, the horizontal wind direction defines the north wind as 0°, and the degree increases in the clockwise direction.

Similar to the results observed in the horizontal verification experiment, a long-live stratified structure can be seen at night from the horizontal wind speed (Fig. 4(b)). And the wind speed has an obvious increase with height in the stable ABL due to the LLJ, so the corresponding horizontal wind shear around it is also larger at night. Usually, the atmosphere layer has a wider and thinner structure at this time result from the stable atmosphere stratification. After the sunrise, with the air is warmed and the ground is unevenly heated, convection activities begin to occur. It can be verified from the vertical wind speed, differs

from the stable state that approaches zero most of the time at night, it changes fast between positive and negative numbers (negative means upward). The stable structure and LLJ that occurred at night are also gradually vanished, as shown in the horizontal wind speed and wind direction, the variances obviously decrease in the vertical direction during the day. Correspondingly, the calculated horizontal wind shear is smaller compared with night.

The results of temperature and pressure profiles in 0-5 km derived from MWR are shown in Fig. 4(d), (h). Obviously diurnal

variations can be seen from the continuous result of the temperature profile. It should be mentioned that because MWR does not have the pressure profile data, considering the atmospheric pressure normally decreases monotonously and rapidly with height, we use the barometric formula (Wallace, 1977) to calculate the pressure profile under the assumption of the temperature gradient is a constant within a certain height range. The pressure in the height of $h_i$ can be derived through:

$$P_i = P_{i-1}\left(1 - \frac{k(h_i - h_{i-1})}{T_{i-1}}\right)^{\frac{gM}{Rk}},$$ (10)

where $T_{i-1}$ and $P_{i-1}$ are the temperature (K) and pressure (hPa) in $h_{i-1}$, respectively. The $k$ is the temperature gradient (K/m) between the height of $h_i$ and $h_{i-1}$, $g$ is the gravitational acceleration (9.8 m/s$^2$), $M$ is the molar mass of Earth's air (0.029 kg/mol), R is the universal gas constant (8.314 J/(mol·K)). Using the temperature profile measured by MWR and the recorded surface pressure, the pressure profile can be calculated from the bottom (the experimental site is 1160 m above sea level) to the top through the differential method.

Since the distance resolution of MWR is lower than CDWL, to keep the original trend of the temperature and pressure profiles and reduce the fluctuation as much as possible, the original temperature profile is interpolated according to the CDWL resolution firstly. Then the temperature gradient is calculated by the linear fit in a moving height range of 200 m. The temperature and temperature gradient profiles at different times acquired from MWR on September 06-07 are respectively shown in Fig. 5(a)-(b) and (e)-(f).

It can be seen from the diurnal and nocturnal temperature profiles that the temperature trends are smoother and more regular compared with wind field results. Therefore, the temperature gradient profiles calculated after processing can not only reflect the overall variation trend, but also reduce the error caused by the jitter of the instrument. In addition, it can be found from the





temperature gradient profiles that the terrain here is not as prone to form a strong temperature inversion layer as in cities. In regard to the pressure profiles, since the pressure decreases faster with increasing height and its variation is more stable (surface recorded data are generally within 10 hPa throughout the day), the results of the profiles at the different moments are quite close (Fig. 5(c)-(d)). A similar characteristic can be found in the gradient profiles (Fig. 5(g)-(h)).

According to the wind field information derived from CDWL and meteorological data recorded by MWR, the two days continuous observation results of $C_n^2$ profile can be retrieved in Fig. 6(d). Compared with the TKEDR profile (Fig. 4(g)), it is obvious that there are similar trends between them most of the time because they are both key parameters to describe the intensity of turbulence. Nevertheless, the $C_n^2$ profile can reveal more details in time and space scale because it is also related to wind shear and temperature gradient.

During the night-time, the data are more continuous and stratified structures can also be found near the strong wind shear layer (00:00-07:00, September 07). Besides, during the period around 01:00-04:00, with the growing intensity of wind shear generated by LLJ, activities in the vertical direction increases, which causes the $C_n^2$ rise in the night. In contrast, there is a lot of fluctuation during the daytime. However, different with the disorderly results in vertical wind speed and wind shear (Fig. 4(e)-(f)), the $C_n^2$ profile have more structures similar to the turbulent eddies. The eddies are obviously larger and denser near the surface due to stronger convective activities, then with the increase of height, they become smaller and sparser. It would be more stable like the night-time and clearly decrease with height after a time average of the results, but considering the rapidly changing atmospheric environment during the day, it can more accurately reflect the real atmospheric conditions in this time resolution.

### 3.4 Results analysis

To analysis the turbulence characteristics in a more comprehensive way, several key parameters are shown in Fig. 6. In the profile of the potential temperature (Fig. 6(a)), it is increasing with height most of the time. Besides, the Brunt-Väisälä frequency squared (Fig. 6(b)) is positive and has a stratified structure most of the time. Combining these features, we can assume that the atmosphere stratification is stable overall. The Richardson number is also an important parameter for rough predicting of air turbulence (Banakh and Smalikho, 2018; Banakh et al., 2020b; Venayagamoorthy and Koseff, 2016). It is a dimensionless number that considering both the effects of the buoyancy term and wind shear term. It has the expression of:

$$R_i = \frac{N^2}{S^2} = \frac{g}{\theta} \frac{\frac{\partial \theta}{\partial z}}{\left(\frac{\partial u}{\partial z}\right)^2 + \left(\frac{\partial v}{\partial z}\right)^2} , \qquad (11)$$

where the meaning of these characters is the same as in Eq. (5) and Eq. (9). If $N^2 < S^2$, the turbulence is strengthened due to the domination of wind shear. If $N^2 > S^2$, the turbulence is suppressed by the buoyancy term. There is a critical Richardson number ($R_c$) of 0.25 to refer to (Miles, 1961; Stull, 1998). When $R_i < R_c$, it indicates the small-scale perturbation of turbulence. It needs to mention that the $R_i$ is used in stably stratified turbulence, and it becomes much less meaningful when $N^2 < 0$. So, in the calculation of $R_i$ in this paper using the "bubble sort" algorithm proposed by Thorpe (Thorpe, 1977) firstly to re-sort the





potential temperature in a monotonically increasing order. The result of $R_i$ is shown in Fig. 6(c). Similar to the $N^2$, it also has

the stratified structure and increases after the night falls, which shows a more stable stratification. Significantly, the overall distribution features are quite consistent with $C_n^2$.

Figure 6(e)-(g) are the perturbations of temperature, horizontal wind speed, and vertical wind speed, respectively. They are derived from the following procedures. Firstly, the mean background of these parameters is calculated by a moving average with a window of 1 h. Secondly, the original perturbation is the difference between raw data and mean background. Thirdly,

to reduce high-frequency noises, the original perturbation is smoothed by a moving average of three points. To analyse the wave structures more intuitively, the horizontal and vertical wind speeds within the height of 1-2 km are averaged, and then using the procedures above to get the perturbations (Fig. 6(h)).

In the result of $C_n^2$ (Fig. 6(d)), stratified structures similar to $N^2$ and $R_i$ can be found, especially on Sep. 06. But unlike most of the time, $C_n^2$ has a decreasing layer of around 1km. Combining the results of $N^2$, there are two layers with inversion of the

density that can be observed around this height. Therefore, the decrease of $C_n^2$ may due to the relatively stable atmospheric stratification between these two layers.

In addition, there are vertical structures in the daytime of $C_n^2$ on Sep. 6. By comparing with Fig. 6(e)-(h), we can find that they are closely related to the wave structures of $T'$, $U'$, and $w'$. Unlike the gravity waves, these waves are more likely driven by the cloud's blocking of solar radiation, which causing temperature fluctuation. When $T' < 0$, then the $w' > 0$, indicating the

atmosphere is depositional, on the contrary, the atmosphere is uplifting. These phenomena generating the perturbation of TKEDR directly, which in turn effecting the value of $C_n^2$.

In Fig. 4(b), an LLJ with a maximum speed of around 14 m/s is forming below 1 km after 21:00 on the 6th, which leads to the increase of wind shear and $C_n^2$ around it, and also the perturbation of temperature. However, what is interesting is that although the wind shear is still high after 03:00, and the $R_i < R_c$, but a large value of $N^2$ indicates the stratification is quite stable. So,

the buoyancy term suppresses the generation of convective activity (vertical wind perturbation is near to zero) and resulting in the reduction of $C_n^2$. Besides, in the area above 1km from 9:00-12:00 on the 7th, the discrepancy between $C_n^2$ and $R_i$ can also be better understood by the stationary $w'$.

To observe the different $C_n^2$ profile characteristics during the day and night, from 15:00 on the 06th to 15:00 on the 07th, five sets of continuous profiles are selected every 6 hours for analysis. As shown in Figure 7, each group has six consecutive

profiles for a total period of around 10 minutes. The raw data in original distance resolution are depicted in black dashed lines, the blue and red lines are the results after 100 m and 200 m moving average in the range of 0-2.2 km and >2.2 km of the original $C_n^2$ profiles, respectively. Besides, results drawn in the second and third-row represent the profiles of night, which are represented with blue lines, and the remaining rows are the profiles of day, which are depicted as red lines.

To compare with, the Hufnagel/Andrews/Phillips (HAP) model (Andrews et al., 2010, 2012) is drawn as green dot-dash lines

in Fig. 7. The model is a modification of the HV model that takes into account the power-law relationship with height near the ground. It can be calculated by:

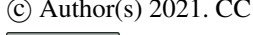



$$C_n^2(h) = M \left[ 0.00594 (\tfrac{w}{27})^2 \left( \tfrac{h+h_s}{10^5} \right)^{10} e^{-\frac{h+h_s}{1000}} + 2.7 \times 10^{-16} e^{-\frac{h+h_s}{1500}} \right] + C_n^2(h_0) \left( \tfrac{h_0}{h} \right)^p . \tag{12}$$

where $M$ is a scaling factor that is related to local characteristics ($M = 1$ is applied in this paper), $h_s$ is the height above the sea level, $w$ is the average wind speed in high-altitude (normally 21 m/s), $h_0$ and $C_n^2(h_0)$ are the height above the ground of

280 the instrument and the measured $C_n^2$, respectively. Power law $p$ typically takes 2/3 at night, and it varies around 4/3 depending on the time during the day.

Firstly, it is not hard to find that the profiles at adjacent moments, especially at night, have a strong correlation and continuity. Secondly, through observe the colored lines, most of the results at night are one to two orders of magnitude lower than that in the daytime at the same height. Similar to the HAP model, the intensity of $C_n^2$ decreases distinctly with height, and it drops

faster during the day than at night near the ground, especially within 1 km. Considering the model is the averaged profile, it is reasonable that the results fluctuate near the profile most of the time. As for the differences in the fourth raw, it is mainly due to the boundary layer is still being uplifted at this moment. In the meantime, the development process of turbulence at different heights can be observed via the high temporal and spatial resolution results of raw data. During the day, with the CBL being lifted, there is obviously stronger fluctuation both in time and space, which is also being found in the actual observations by

most radiosonde methods (Barletti et al., 1974; Martini et al., 2017).

In the order of time series, the mean intensity of $C_n^2$ is the highest around 15:00. After the sunset, the $C_n^2$ decreases to the minimum near 03:00, and the profile is also the most stable at this moment. Around 09:00 in the morning, $C_n^2$ grows from the surface at first but the upper region still remains low. Until noon, the profile becomes unstable with the $C_n^2$ reaches to the maximum again. All these characteristics reflect the credibility of the results.

## 295 4 Conclusions and discussion

To analyse the atmospheric turbulence in high resolution, we proposed a new method according to Tartarski's theory. A contrastive experiment was conducted horizontally with LAS to verify the method. Based on the result, we obtained the continuous $C_n^2$ profile with high temporal and spatial resolution simultaneously by combining the advantages of CDWL and MWR. It is significant for studying the complex and fast-changing atmospheric environment.

We discovered the stable stratified structure at night through the analysis of the buoyancy term and Richardson number. The process of how turbulence developed from the surface and gradually decreasing upwards during the day. And the wave structures are found in the daytime due to the perturbation of temperature and wind speed. Next, through observing the continuous typical profiles, the stability of the system could be found. And the rationality of the results was verified by comparing them with the HAP turbulence profile model.

When retrieving the vertical profile, we assumed most of the time in the region below the CBL basically satisfies the Kolmogorov "2/3 law" (Tatarskii, 1961). However, is it still suitable when dealing with the region above CBL or during the transition period when anisotropic turbulence is more likely to occur due to the stratification stabilizing (Banakh and Smalikho,



2019) needs to be further verified. In addition, due to the scales of turbulent eddies are changing with heights, to obtain a more accurate profile, it is necessary to find and adjust the range resolution of CDWL and MRW more appropriately in future work.

*Data availability.* All the data mentioned in the paper can be downloaded from https://figshare.com/articles/dataset/Turbulence_detection_experiments/16607306.

*Author Contributions.* Conceptualization, H.Y.X. and P.J.; methodology, P.J. and L.W.; experiment, J.L.Y. and P.J.; data curation, L.W., J.L.Y., P.J., K.N.W.; formal analysis, P.J. and L.W.; writing—original draft preparation, P.J.; writing—review and editing, P.J., K.N.W., L.W., J.L.Y. and H.Y.X.; visualization, P.J. and H.Y.X.; supervision, H.Y.X.; project administration,
H.Y.X. All authors have read and agreed to the published version of the manuscript.

*Conflicts of Interest.* The authors declare no conflict of interest.

*Acknowledgments.* Thanks to Dr. Hao Liu from atmospheric boundary layer physics laboratory of USTC for providing the data of wind tower.

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


**Table 1. Key Parameters of the CDWL.**

| Parameter | Value |
|---|---|
| Wavelength | 1548 nm |
| Pulse energy | 300 μJ |
| Pulse width | 200 ns |
| Repetition frequency | 10 kHz |
| Temporal resolution | 2 s |
| Azimuth scanning range | 0 - 360° |
| Zenith scanning range | 0 - 90° |

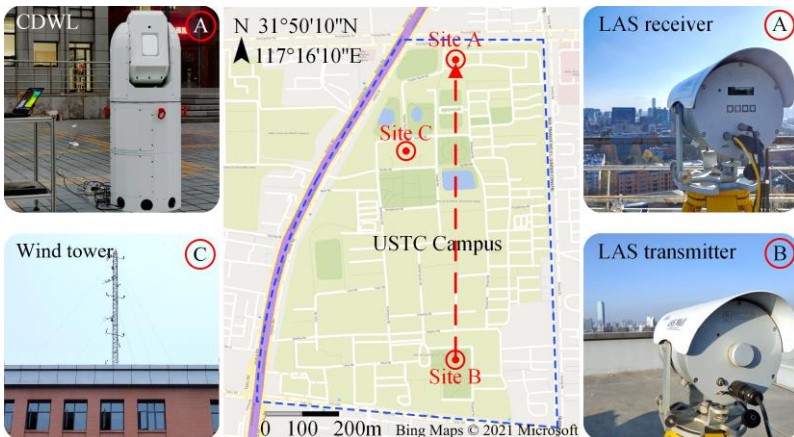

**Figure 1. Instruments layout in the horizontal experiment (USTC, Hefei city).**



**Figure 2. The horizontal wind speed (a), wind direction (b), vertical wind speed (c), wind shear (d), TKEDR (e), temperature and temperature gradient (f), $C_n^2$ and Brunt-Väisälä frequency squared (g) retrieved from CDWL, wind tower and LAS in the observations from September 26 to October 01, 2020, local time.**



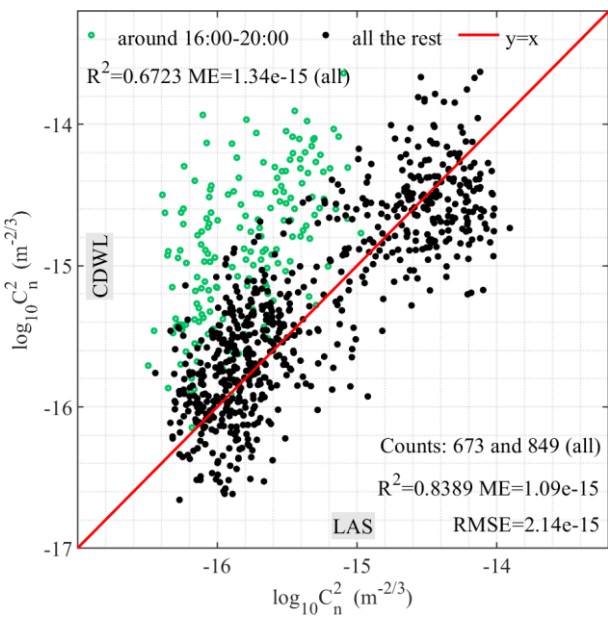

**Figure 3. Comparative statistical analysis of LAS and CDWL observation results from September 26 to October 01, 2020, local time.**

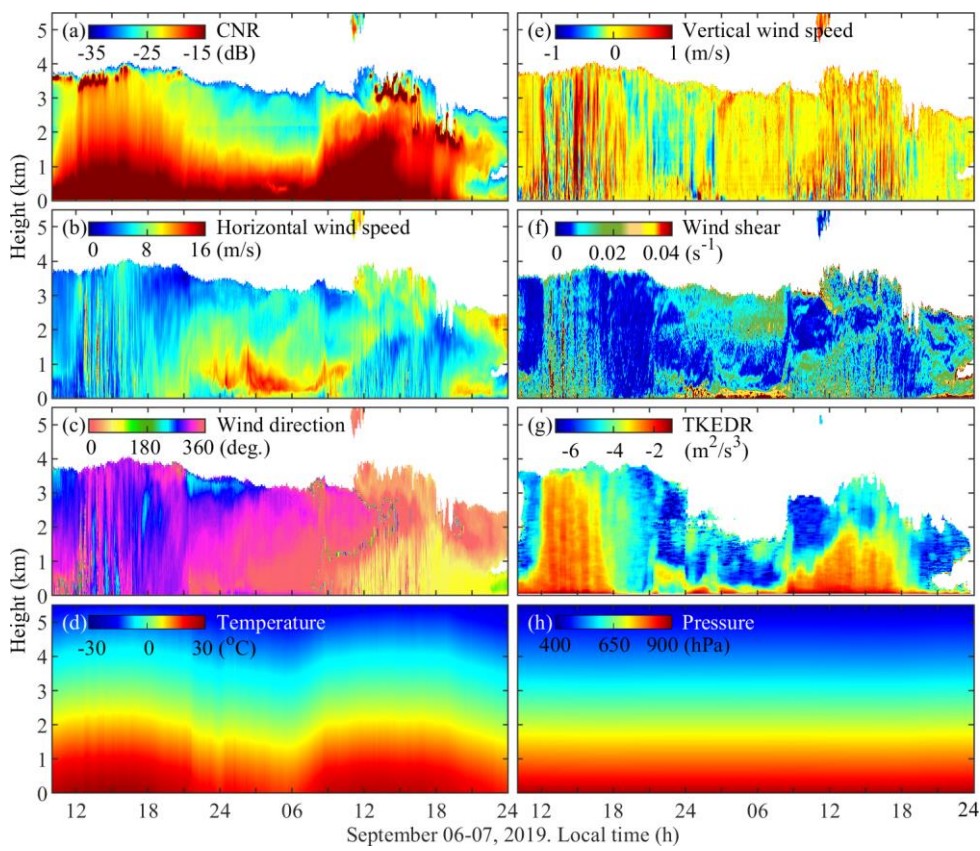




**Figure 4. The CNR (a), horizontal wind speed (b), wind direction (c), temperature (d), vertical wind speed (e), wind shear (f), TKEDR**
**(g), and the pressure (h) derived from CDWL and MWR in the observations from September 06 to September 07, 2019, local time.**

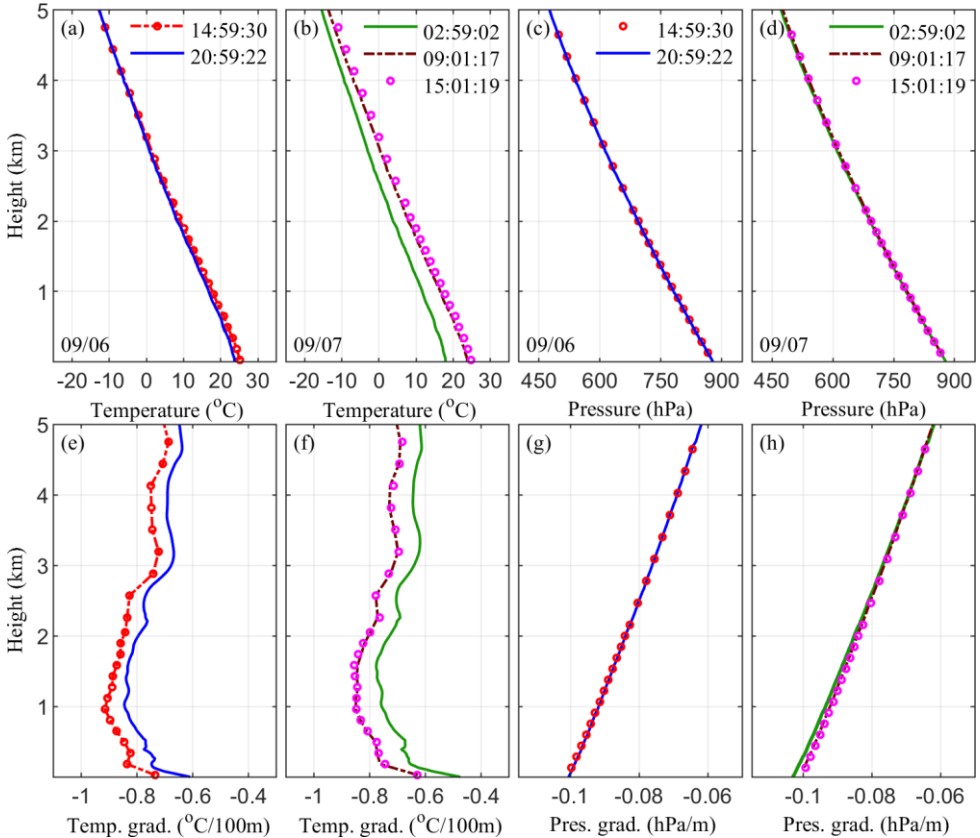

**Figure 5. The results of temperature profiles (a)-(b), temperature gradient profiles (e)-(f), pressure profiles (c)-(d), and pressure gradient profile (g)-(h) derived from MWR at different time on September 06-07, 2019, local time.**







**Figure 6. The potential temperature (a), Brunt-Väisälä frequency squared (b), Richardson number (c), $C_n^2$ (d) profiles, perturbations of temperature (e), horizontal wind speed (f), vertical wind speed (g), and average wind speed perturbations within 1-2 km (h) derived from CDWL and MWR in the observations from September 06 to September 07, 2019, local time.**

**Figure 7. Profiles of $C_n^2$ in raw data, distance moving averaged and calculated from the HAP model during the period of 14:59:30-15:11:49 (a1)-(a6), 20:59:22-21:11:41 (b1)-(b6) on September 06, 02:59:02-03:11:21 (c1)-(c6), 09:01:17-09:13:36 (d1)-(d6), 15:01:19-15:13:40 (e1)-(e6) on September 07, 2019, local time.**
