# Peer review of "Turbulence Detection in the Atmospheric Boundary Layer using Coherent Doppler Wind Lidar and Microwave Radiometer"

_Atmospheric Measurement Techniques, 2021_

## Referee Comment (RC1)

**Comments to the manuscript "Turbulence Detection in the Atmospheric Boundary Layer using Coherent Doppler Wind Lidar and Microwave Radiometer" by Pu Jiang, Jinlong Yuan, Kenan Wu, Lu Wang, and Haiyun Xia**

Manuscript presents the experimental results intending on demonstration of the possibility of vertical profiling the refractive index structure constant based on estimation of the turbulent kinetic energy dissipation rate and gradients of wind velocity and potential temperature from data of wind coherent lidar and microwave radiometer. For determining the refractive index structure constant, Eq. (4) in the manuscript is used. Eq. (4) follows from the formulae listed in Tatarskii, 1961 (see References in the manuscript).

The main remark to the manuscript is following. V.I. Tatarskii wrote in Tatarskii, 1961, that these formulae are true for the surface layer of the atmosphere. As to the heights above the surface layer, he noted, a lot of experiments are required to test applicability of these formulae. Actually, as it follows from the ground experiment described in the manuscript, Eq.(4) does not work even in the surface layer under the stable conditions, it gives the refractive index structure constant values which differ from the scintillometer results by two orders. The temperature stratification in the atmosphere is stable one at the heights exceeding the boundary layer height independently on the stratification in the boundary layer. Thus, Eq. (4) does not work over the boundary layer in any case.

The rest remarks are following.

1) Lines 15-16: "..... the mean error and standard deviation is $1.09 \times 10^{-15}$ m$^{-2/3}$ and $2.14 \times 10^{-15}$ m$^{-2/3}$, respectively."

That says about nothing. Relative units are more informative.

2) Line 70. Eq. (1) is listed in Tatarskii, 1961, for the temperature structure constant. Relation between the refractive index structure constant and the temperature one is commonly known.

3) Lines 108-109: " In the vertical direction of 0-2.17 km, 2.17-4.76 km, and 4.76-11.26 km, the range resolution is 26 m, 52 m, and 130 m, respectively."

Pulsed lidars have dead zone, diapason "0-2.17 km" is not true. The same is for Figs. 2,4,6.

In Fig. 2e, instead of "-8   -2" should be "$10^{-8}$   $10^{-2}$". The same is for Figs.4g, 6d.

4) Line 122:

What means " the DAVIS weather station"?

5) Lines 123,129: "The receiving and transmitting ends of LAS are located at the height of 55 m at site A and site B respectively. ".... "temperature data recorded at the height of 2m, 8m, and 18m"

Difference in heights leads to difference in the refractive index structure constant about three times. The structure constant decreases with height.

6) Figs. 2f,5 say about nothing. The potential temperature and its gradient should be instead of temperature and temperature gradient to see the temperature stratification and its variations with height.

7) The Richardson number in Fig. 6c is positive. That means, during measurements there was stable temperature stratification in the atmosphere. The applicability of Eq.(4) in such conditions is under question. As well as correctness of the profiles in Fig.7. Figs. 2g, 3 (green dots) demonstrate that at stable conditions there is large difference between the results of the scintillometer and calculations based on Eq. (4).

---

## Author Comment (AC1)

**Dear Referee #1:**

Thank you for your valuable comments and suggestions which helped us significantly to improve our manuscript. We have considered all comments carefully and revised the manuscript. Our point-by-point responses to all your comments are listed below in blue fonts and the changes in the manuscript are listed in *blue italic fonts*.

**Anonymous Referee #1:**

Manuscript presents the experimental results intending on demonstration of the possibility of vertical profiling the refractive index structure constant based on estimation of the turbulent kinetic energy dissipation rate and gradients of wind velocity and potential temperature from data of wind coherent lidar and microwave radiometer. For determining the refractive index structure constant, Eq. (4) in the manuscript is used. Eq. (4) follows from the formulae listed in Tatarskii, 1961 (see References in the manuscript).

**General Comments:**

The main remark to the manuscript is following. V.I. Tatarskii wrote in Tatarskii, 1961, that these formulae are true for the surface layer of the atmosphere. As to the heights above the surface layer, he noted, a lot of experiments are required to test applicability of these formulae. Actually, as it follows from the ground experiment described in the manuscript, Eq.(4) does not work even in the surface layer under the stable conditions, it gives the refractive index structure constant values which differ from the scintillometer results by two orders. The temperature stratification in the atmosphere is stable one at the heights exceeding the boundary layer height independently on the stratification in the boundary layer. Thus, Eq. (4) does not work over the boundary layer in any case.

**Response:** Firstly, thanks for your comments about the limitation of Eq. (4) under different circumstances. Based on our horizontal experiment, the reviewer points out that Eq. (4) doesn't work under stable conditions even in the ground, then he/she mentions this method is limited in the atmospheric boundary layer (ABL).

Actually, in the ground experiment, the discrepancy between the method using Eq. (4) and the scintillometer mainly within the transition period around 16:00-20:00 as shown in Fig. 2(g). When the stratification structure becomes stable at night, the TKEDR also gradually decreases, so the $C_n^2$ obtained from them becomes consistent again. However, the results in the night coincide not as well as in the daytime. To study the limitation of this method, we added the uncertainty analyses in the revised manuscript by calculating the relative error of $C_n^2$ and the integral scale of turbulence.

In the horizontal results, we find that the integral scale of turbulence $L_v$ drops to the scale smaller than the length used to calculate the TKEDR during the transition period around 16:00-20:00 (Fig. 3(b) in the revised manuscript). This verified the difference between the two instruments is mainly due to the state of the atmosphere changing from isotropic to anisotropic. In the vertical profiles, the $L_v$ grows when the TKEDR decreases with height, which causes the larger relative error of the estimation of $C_n^2$ in the high altitude (Fig. 8 in the revised manuscript).

According to the point of the reviewer, we tested the limits of this method and find the rationality within the ABL, especially within the convective boundary layer (Fig. 7 in the revised manuscript). Thanks again for your comments to make this manuscript more convincing and practical.

**Specific Comments:**

**Comments 1:** Lines 15-16: "... the mean error and standard deviation is 1.09×10-15 m-2/3 and 2.14×10-15 m- 2/3,

respectively."

That says about nothing. Relative units are more informative.

**Response 1:** Thanks for your suggestion, we have added the relative error in the revised Fig. 3(c). It should be mentioned that due to the value of the $C_n^2$ being very small and normally changing between 2-3 orders of magnitude, the calculated relative error could be quite large even a small difference. For example, $\left(2 \times 10^{-15}m^{-\frac{2}{3}} - 1 \times 10^{-15}m^{-\frac{2}{3}}\right)/(1 \times 10^{-15}m^{-\frac{2}{3}}) = 100\%$. So, the relative error is calculated on a logarithmic scale.

**Changes 1:** *Line 179: When using all data for analysis, the correlation coefficient, mean error, and relative error between the two methods are 0.6723, 1.34×10⁻¹⁵ m⁻²/³, and 2.83%, respectively. When using the black dots, the correlation coefficient, mean error, and relative error are 0.8389, 1.09×10⁻¹⁵ m⁻²/³, and 2.04%, respectively.*

**Comments 2:** Line 70. Eq. (1) is listed in Tatarskii, 1961, for the temperature structure constant. Relation between the refractive index structure constant and the temperature one is commonly known.

**Response 2:** Indeed, the method using the relationship between the $C_n^2$ and $C_T^2$ is a common way to estimate the refractive index structure constant. As we discussed in the introduction, most of them acquire the $C_n^2$ profiles through the sounding balloon with a Radiosonde. It normally takes a long time to obtain one profile. However, considering the fast-changing turbulence environment, our purpose and innovation are to seek a method that can detect the turbulence profiles with high temporal and spatial resolution at the same time. Therefore, we use Eq. (4), which contains the dynamics and thermodynamics part, to estimate the $C_n^2$ profiles.

**Comments 3:** Lines 108-109: " In the vertical direction of 0-2.17 km, 2.17-4.76 km, and 4.76-11.26 km, the range resolution is 26 m, 52 m, and 130 m, respectively."

Pulsed lidars have dead zone, diapason "0-2.17 km" is not true. The same is for Figs. 2,4,6.

In Fig. 2e, instead of "-8 -2" should be "10-8 10-2". The same is for Figs.4g, 6d.

**Response 3:** Thanks for your reminder, the lidar we used has a pulse width of 200 ns, which has a blind zone of around 30 m. Now we have corrected the expression and the data in the figures are plotted from 51.96 m (first bin at 60 m and the elevation is 60 degrees).

The value of TKEDR ("-8 -2") are the units in log scale to simplify the expression and the same with Fig. 4(g), 6(d). And we have added the "$\log_{10}()$" in these figures for convenience.

**Changes 3:** *Line 115: The lidar has a pulse width of 200 ns, which has a blind zone of around 30 m. So, in the vertical direction of 0.03-2.20 km, 2.20-4.79 km, and 4.79-11.29 km, the height resolution is 26 m, 52 m, and 130 m, respectively.*

**Comments 4:** What means " the DAVIS weather station"?

**Response 4:** It means the weather station of model DAVIS6162: Wireless Vantage Pro2 Plus. We have added this information to the manuscript.

**Changes 4:** *Line 130: the weather station (DAVIS6162: Wireless Vantage Pro2 Plus).*

**Comments 5:** Line Lines 123,129: "The receiving and transmitting ends of LAS are located at the height of 55 m at site A and site B respectively. "… "temperature data recorded at the height of 2m, 8m, and 18m"

Difference in heights leads to difference in the refractive index structure constant about three times. The structure constant decreases with height.

**Response 5:** Thanks for reminding, actually the temperature data were recorded at the height of 2m, 8m, and 18m of the wind tower which was placed on the top of a 6-story building about 30 meters high. So, the height difference is quite small and now we explained in the paper.

**Changes 5:** *Line 134: The wind tower is placed on the top of a 6-story building about 30 meters high at site C to record the continuous data of temperature and for the calculation of temperature gradient.*

**Comments 6:** Figs. 2f, 5 say about nothing. The potential temperature and its gradient should be instead of temperature and temperature gradient to see the temperature stratification and its variations with height.

**Response 6:** Thanks for your advice, the temperature and its gradient were plotted in Fig. 2(f) to discover the temperature inversion phenomenon and the strong negative correlation with the $C_n^2$. And the Brunt-Väisälä frequency squared $N^2$ in Fig. 2(g) was drawn to reflect the temperature stratification structure. As shown in the green line in the following: the potential temperature gradient has a similar trend with the temperature gradient, especially with the $N^2$. So, to avoid giving redundant information and reveal the negative correlation between temperature inversion and the $C_n^2$, we kept the temperature gradient result in the horizontal experiment.

[Figure]

Besides, we have added the potential temperature and its gradient profiles to see the temperature stratification in Fig. 5 in the revised manuscript according to your suggestion.

**Changes 6:**

[Figure]

*Figure 5. The results of temperature profiles (a)-(b), temperature gradient profiles (e)-(f), potential temperature profiles (c)-(d), and potential temperature gradient profiles (g)-(h) derived from MWR and the barometric formula at different times on September 06-07, 2019, local time.*

**Comments 7:** The Richardson number in Fig. 6c is positive. That means, during measurements there was stable temperature stratification in the atmosphere. The applicability of Eq. (4) in such conditions is under question. As well as correctness of the profiles in Fig.7. Figs. 2g, 3 (green dots) demonstrate that at stable conditions there is large difference between the results of the scintillometer and calculations based on Eq. (4).

**Response 7:** Firstly, the Richardson number was calculated using the "bubble sort" algorithm proposed by Thorpe (Thorpe, 1977) to re-sort the potential temperature in a monotonically increasing order, which caused the positive value of the Richardson number. From Eq. 11, one can see that the sign of the $R_i$ should be the same as $N^2$. Secondly, the $R_i$ is the ratio of $N^2$ to wind shear $S^2$. When $0 < R_i < 1$, turbulence is easy to occur due to the domination of wind shear. So, the positive $R_i$ doesn't mean a stable layer, and a more specific illustration about $R_i$ can be found in Line 281 in the revised manuscript. Thirdly, the differences in Fig. 2(g), 3 (green dots) have been explained in the general comments, which are mainly within the transition period around 16:00-20:00 rather than all of the stable conditions at night. Moreover, the green lines in Fig. 7 are the HAP turbulence model that takes into account the power-law relationship with height near the ground. So, they are drawn here mainly to compare the surface layer and the model cannot represent a specific local feature. Finally, we have supplemented the analysis of the applicability and uncertainty of this method under different circumstances in the revised manuscript.

**Changes 7:** There are several changes related to the analysis of the limitation of this method. Parts of them are listed in the following:

*Figure 3. The relative error of the estimation of TKEDR and $C_n^2$ (a), integral scale $L_v$ (b), and comparative statistical analysis of LAS and CDWL observation results from September 26 to October 01, 2020, local time.*

[Figure]

*Figure 7. The relative error of the estimation of TKEDR (a), $C_n^2$ (b), and integral scale $L_v$ (c) calculated from CDWL and MWR in the observations from September 06 to September 07, 2019, local time.*

*Line 315: Then, the relative error of estimation of TKEDR, $C_n^2$ and the integral scale $L_v$ are calculated vertically in Fig. 7. The region with a relative error greater than 50% are marked in light yellow in Fig. 7(a) and (b). From the results, it can be seen that when using this method to obtain the profiles, a small relative error ($RE_{C_n^2}$ is mostly within 30%) can be maintained in the ABL, especially in the CBL. In the meantime, $L_v$ is basically under 1 km in this area, so that $R' > L_v$ is satisfied, which means a low $RE_{TKEDR}$. After 18:00, with the height of the boundary layer decreases, the TKEDR drops rapidly at high altitude, and the $L_v$ becomes larger than 2 km. As a result, the calculated $RE_{TKEDR}$ and $RE_{C_n^2}$ also grow as shown in the figure. During the period of the atmosphere changes from convection to laminar flow (around 18:00 to 21:00), a sudden increase in relative error can be found similar to the horizontal experiment. After the atmosphere stabilized at night, the relative error of TKEDR and $C_n^2$ begin to gradually decrease, but mainly within the mixing layer.*

Best regards!
Sincerely yours,
Haiyun Xia
School of Earth and Space Sciences,
University of Science and Technology of China.
96 Jinzhai rd. Hefei, Anhui, CHINA, 230026.

---

## Author Comment (AC2)

**Dear Referee #2:**

Thank you for your valuable comments and suggestions which helped us significantly to improve our manuscript. We have considered all comments carefully and revised the manuscript. Our point-by-point responses to all your comments are listed below in blue fonts and the changes in the manuscript are listed in *blue italic fonts*.

**Anonymous Referee #2:**

Pu et al. present an interesting study, which uses a combination of instruments which are increasingly used in operational observation of the ABL, i.e. Doppler wind lidar and microwave radiometer. They derive the structure parameter Cn2 from profile measurements, which is quite uncommon in boundary-layer meteorology profiling, but can be reasonable for the applications in optics and astronomy. Despite this novel approach the authors fail to convincingly show that their method really provides data that is valid and useful for the described applications. No uncertainty estimation is presented and confronted with the requirements. For these reasons, I cannot recommend the manuscript for publication in AMT unless major revisions are implemented.

**Response:** Thank you for your recognition of the novelty in the manuscript. The purpose of our work is to obtain the turbulence profiles with high temporal and spatial resolution simultaneously. To make our results more convincing, we took the verification experiments horizontally with the scintillometer due to the difficulty of vertical experiments with high resolution. However, we become to realize the significance of results reliability evaluation thanks to your reminder. Now we have added the uncertainty analyses by calculating the relative error of TKEDR, $C_n^2$ and the integral scale of turbulence in both horizontal and vertical experiments in the revised manuscript.

**Changes:** *Line 368: Through the calculation of the relative error and integral scale of turbulence, we analyzed the uncertainty and limitation when using this method at different times and altitudes. In the horizontal results, the $C_n^2$ retrieved by CDWL in the night (especially the transition period) coincides not as well as in the daytime when the integral scale of turbulence $L_v$ reduces to a value smaller than $m\Delta y$. In the vertical profiles, the $C_n^2$ can be estimated with a low relative error under the ABL. And the $L_v$ grows when the TKEDR decreases with height, which leads to a larger $RE_{C_n^2}$ in the high altitude.*

**General Comments:**

Doppler wind lidar turbulence retrievals are always problematic in low turbulence regimes, because the volume averaging effect can only be corrected to a certain limit and small-scale turbulence cannot be captured. The cited work by Smalikho gives clear boundaries and criteria under which dissipation rates can be obtained with a reasonable uncertainty. It mostly depends on the integral length scale of turbulence. This should be considered in this study as well.

Microwave radiometers are known to not be able to capture strong temperature gradients very well. This can be problematic at the tropopause, but also in nighttime inversion layers and at the top of the boundary layer. However, turbulence can particularly occur at these levels and Cn2 should be strongly affected. The authors do not discuss this and the implications on the accuracy of their retrievals.

English language should be somewhat improved in the next revision. Some paragraphs are hard to understand.

**Response:** Thanks for your constructive suggestions. Indeed, small-scale turbulence is hard to be found by

Doppler wind lidar, especially at the high altitude in the nighttime. The criteria using the integral length scale of turbulence to estimate the uncertainty given by Smalikho is an effective way, and we have supplemented this part in the revised manuscript as we answered above.

Compare with the wind tower or radiosonde, the microwave radiometer (MWR) uses a remote sensing method to gain the temperature profiles. The MWR is indeed hard to capture strong temperature gradients layers, such as the tropopause, inversion layers, or the top of the boundary layer as the reviewer mentioned. However, these areas normally exist quite strong turbulence activities, which are vital in atmospheric turbulence research, especially in the boundary layer theory. Consequently, we have discussed this issue in the revised version.

Finally, we checked our revised manuscript and improved the grammar and sentences to make them more understandable. Thanks again for all your advice.

**Changes:** There are several changes related to the uncertainty analysis in the revised manuscript. The main principle is listed in the following:

*Line 180-210: To testify the applicability and its uncertainty of this method under different circumstances, the relative error of the estimation of $C_n^2$ ($RE_{C_n^2}$) is calculated according to Eq. (4) as follows:*

$$RE_{C_n^2} = \left[\frac{4}{9}(RE_{TKEDR})^2 + 4(RE_{windshear})^2 + 4\left(RE_{M(T,P)}\right)^2\right]^{\frac{1}{2}} , \tag{10}$$

*where the $RE_{TKEDR}$ is estimated based on the lidar system parameters, the value of TKEDR, and the instrumental error of the radial velocity ($\sigma_e$) (Banakh et al., 2017; Smalikho and Banakh, 2017). The $\sigma_e$ is mainly affected by the CNR and it is calculated by the model of Cramer-Rao lower bound (CRLB) with an assumption of a Gaussian laser pulse (Frehlich et al., 1994; Rye and Hardesty, 1993a, b). $RE_{windshear}$ can be derived from the sum of relative error of horizontal wind in different altitudes divided by the distance between two layers. According to Eq. (8), the $RE_{M(T,P)}$ is estimated by the relative error of temperature ($RE_T$) and pressure ($RE_P$). In this paper, $RE_T$ takes $\pm 1\,K$ @ 280 K and $RE_P$ takes $\pm 1\,hPa$ @ 800 hPa.*

*The six-day continuous results of $RE_{TKEDR}$ and $RE_{C_n^2}$ are plotted in Fig. 3(a). Since the height set in the horizontal experiment is near the surface layer, the CNR is high and the instrumental error of estimation of the wind field is quite small. Therefore, the relative error of TKEDR and $C_n^2$ are within 10% most of the time, which demonstrates the robustness of this method. The $RE_{C_n^2}$ is also shown in Fig. 2(g) with a shaded area error bar. However, since the $C_n^2$ is usually compared on a log scale, a 10% relative error is not obvious in the figure.*

*The integral scale $L_v$ is an indicator reflecting the rationality of the turbulence parameters retrieval (Smalikho and Banakh, 2020). It can be calculated from the radial velocity variance averaged over all azimuth angles ($\bar{\sigma}_r^2$) and TKEDR (Smalikho and Banakh, 2017):*

$$L_v = 0.698(\bar{\sigma}_r^2)^{3/2}/\varepsilon , \tag{11}$$

*where $\bar{\sigma}_r^2$ has the expression of:*

$$\bar{\sigma}_r^2 = (sin\varphi)^2\sigma_w^2 + \left(\frac{1}{2}\right)(cos\varphi)^2(\sigma_u^2 + \sigma_v^2) , \tag{12}$$

*where $\varphi = 60°$, $\sigma_u^2 = <(u')^2>$, $\sigma_v^2 = <(v')^2>$, $\sigma_w^2 = <(w')^2>$ are the variances of the fluctuations of zonal, meridional and vertical wind components. On the one hand, when the distance $m\Delta y < L_v$ is satisfied ($\Delta y$ is the spatial distance between the centers of two neighboring probing volumes, $m\Delta y = 5.24\,m$ in the horizontal experiment), then the local isotropy of turbulence holds. On the other hand, if the radius of the scanning cone at a certain height ($R' = 30\,m$) is comparable with or even smaller than $L_v$, the relative error of estimation of TKEDR becomes larger.*

*From the results of $L_v$ in Fig. 3(b), one can see that it is mainly distributed between the $m\Delta y$ and $R'$, which means this method work and remain a low relative error most of the time during the experiment. During the transition period around 16:00-20:00, the integral scale of turbulence $L_v$ drops to the scale smaller than the $m\Delta y$ used to calculate the TKEDR. Like the results shown in the temperature gradient and $N^2$, this verified the difference between the two instruments are mainly due to the prominent motion state of the atmosphere changes from convective to laminar flow, which means the local turbulence translates from isotropic into anisotropic.*

**Specific Comments:**

**Comments 1:** p.1, l.16: check units

**Response 1:** Thank you for your reminder, we have replaced the standard deviation with the relative error which is more informative.

**Changes 1:** *Line 15: ...the correlation coefficient between them in the six-day observation is 0.8389, the mean and relative error is $1.09\times10^{-15}m^{-2/3}$ and 2.04%, respectively.*

**Comments 2:** p.3, Eq.2: $z$ is missing in the equation

**Response 2:** Thanks a lot. We have corrected the intensity of the wind shear *S* into *S(z)* in the equation.

**Changes 2:** *Line 82: Eq. 2*

$$L_0^{4/3}(z) = \begin{cases} 0.1^{4/3} \times 10^{1.64+42\times S(z)}, troposphere \\ 0.1^{4/3} \times 10^{0.506+50\times S(z)}, stratosphere \end{cases} ,$$  (2)

**Comments 3:** p.4, l.100f: The MWR does not provide pressure profiles.

**Response 3:** Thanks for reminding us. We have fixed it in a more precise way.

**Changes 3:** *Line 107: ...derived from microwave radiometer and the barometric formula.*

**Comments 4:** p.5, l.132f: so, I understand that the temperature gradient is not measured, but Fig.2 says that Cn2 is calculated from wind tower as well.

**Response 4:** Sorry for the misunderstanding, the $C_n^2$ was derived from the CDWL and LAS, instead of the wind tower. Now we have changed the expression.

**Changes 4:** *Figure 2. The horizontal wind speed (a), wind direction (b), vertical wind speed (c), wind shear (d), $log_{10}(TKEDR)$ (e), $log_{10}C_n^2$ and its shaded area error bar (g) retrieved from CDWL, temperature, temperature gradient (f) and Brunt-Väisälä frequency squared (g) recorded by wind tower, $log_{10}C_n^2$ (g, red) obtained from LAS in the observations from September 26 to October 01, 2020, local time.*

**Comments 5:** p.6, l.169ff: I do not think that correlation coefficient and mean error are a good overall estimate here. It should probably be presented in some relation to the turbulence and stability regime.

**Response 5:** According to your suggestion, we have added the uncertainty analysis and explained it more

reasonably in the revised manuscript.

**Changes 5:**

[Figure]

*Figure 3. The relative error of the estimation of TKEDR and $C_n^2$ (a), integral scale $L_v$ (b), and comparative statistical analysis of LAS and CDWL observation results from September 26 to October 01, 2020, local time.*

*Line 208: From the results of $L_v$ in Fig. 3(b), one can see that it is mainly distributed between the $m\Delta y$ and $R'$, which means this method work and remain a low relative error most of the time during the experiment. During the transition period around 16:00-20:00, the integral scale of turbulence $L_v$ drops to the scale smaller than the $m\Delta y$ used to calculate the TKEDR. Like the results shown in the temperature gradient and $N^2$, this verified the difference between the two instruments are mainly due to the prominent motion state of the atmosphere changes from convective to laminar flow, which means the local turbulence translates from isotropic into anisotropic.*

**Comments 6:** p.8, l.212: I think this is a misinterpretation. I do not think that the smooth profiles reduce the error, but are actually a source of error, as described in the general comments. I am not sure what is meant by "jitter of the instrument" here.

**Response 6:** Actually, we did not smooth the temperature gradient profiles. It was calculated by the moving linear fit in a height range of 200 m. If we obtained the temperature gradient within a short-range, the results would fluctuate widely and become unstable like shown in the wind tower. As you mentioned, the microwave radiometer has a limitation in detecting strong temperature gradients. This phenomenon often occurs at the tropopause, nighttime inversion layers, and the top of the boundary layer. So, if we want to analyze these layers more precisely, a much higher distance resolution is needed for both the MWR and CDWL. Therefore, moving linear fit is adopted when calculating the temperature gradient profiles to avoid the possibility of the wrong estimation. We have explained this in the revised manuscript in case of misunderstanding.

**Changes 6:** *Line 248: The MWR uses a remote sensing method to gain the temperature profiles and it has limitations in detecting strong temperature gradient layers. This phenomenon often occurs at the tropopause, the top of the boundary layer, and the inversion layer, especially at night. In order to analyze these layers more accurately, the distance resolution of both the MWR and CDWL needs to be highly improved. Therefore, the temperature gradient is calculated by the moving linear fit in a height range of 200 m to avoid the possibility of the wrong estimation.*

**Comments 7:** p.8, l.235f: The Richardson number is not a parameter for rough prediction, but a comprehensive turbulence parameter that gives a value for the dynamic stability of the atmosphere. Equation 11 gives the gradient Richardson number, which is a simplification of the flux Richardson number.

**Response 7:** Thank you for your correctness. We have changed the way we describe the Richardson number in the revised manuscript.

**Changes 7:** *Line 281: The Richardson number is also an important and comprehensive turbulence parameter that gives a value for judging the dynamic stability of the atmosphere.*

**Comments 8:** p.20, fig.5: Pressure profile and pressure gradient is not really very interesting here.

**Response 8:** Thanks for your suggestion. The pressure profile and its gradient don't vary a lot with height. So, we changed it into the potential temperature and its gradient to observe the stratification phenomenon in Fig. 5 in the revised manuscript.

**Changes 8:**

[Figure]

*Figure 5. The results of temperature profiles (a)-(b), temperature gradient profiles (e)-(f), potential temperature profiles (c)-(d), and potential temperature gradient profiles (g)-(h) derived from MWR and the barometric formula at different times on September 06-07, 2019, local time.*

Frehlich, R., Hannon, S. M., and Henderson, S. W.: Performance of a 2-µm Coherent Doppler Lidar for Wind Measurements, Journal of Atmospheric and Oceanic Technology, 11, 1517-1528, 10.1175/1520-0426(1994)011<1517:POACDL>2.0.CO;2, 1994.

Rye, B. J., and Hardesty, R. M.: Discrete spectral peak estimation in incoherent backscatter heterodyne lidar. I. Spectral accumulation and the Cramer-Rao lower bound, IEEE Transactions on Geoscience and Remote Sensing, 31, 16-27, 10.1109/36.210440, 1993a.

Rye, B. J., and Hardesty, R. M.: Discrete spectral peak estimation in incoherent backscatter heterodyne lidar. II. Correlogram accumulation, IEEE Transactions on Geoscience and Remote Sensing, 31, 28-35, 10.1109/36.210441, 1993b.

Smalikho, I. N., and Banakh, V. A.: Measurements of wind turbulence parameters by a conically scanning coherent Doppler lidar in the atmospheric boundary layer, Atmos. Meas. Tech., 10, 4191-4208, 10.5194/amt-10-4191-2017, 2017.

Best regards!
Sincerely yours,
Haiyun Xia
School of Earth and Space Sciences,
University of Science and Technology of China.
96 Jinzhai rd. Hefei, Anhui, CHINA, 230026.